# Optimization of Ultrasound-Assisted Cellulase Extraction from *Nymphaea hybrid* Flower and Biological Activities: Antioxidant Activity, Protective Effect against ROS Oxidative Damage in HaCaT Cells and Inhibition of Melanin Production in B16 Cells

**DOI:** 10.3390/molecules27061914

**Published:** 2022-03-16

**Authors:** Hui-Min Liu, Sheng-Nan Lei, Wei Tang, Meng-Han Xun, Zhi-Wei Zhao, Ming-Yan Cheng, Xiao-Dan Zhang, Wei Wang

**Affiliations:** 1School of Perfume & Aroma and Cosmetics, Shanghai Institute of Technology, Shanghai 201418, China; szliuhm@sit.edu.cn (H.-M.L.); 196071222@mail.sit.edu.cn (S.-N.L.); 216071129@mail.sit.edu.cn (W.T.); 206072134@mail.sit.edu.cn (M.-H.X.); 206071246@mail.sit.edu.cn (Z.-W.Z.); 206071241@mail.sit.edu.cn (M.-Y.C.); 2Engineering Research Center of Perfume & Aroma and Cosmetics, Ministry of Education, Shanghai 201418, China; 3Shanghai Shandai Biotechnology Co., Ltd., Qingpu District, Shanghai 201706, China; cncjay@126.com

**Keywords:** *Nymphaea hybrid*, ultrasound-assisted, cellulase, macroporous resins, antioxidant, ROS, melanin

## Abstract

In this study, ultrasonic-assisted cellulase extraction (UCE) was applied to extract flavonoids and polyphenols from the *Nymphaea hybrid* flower. The extraction conditions were optimized using the response surface method (RSM) coupled with a Box-Behnken design. The crude extract of *Nymphaea hybrid* (NHE) was further purified using AB-8 macroporous resins, and the purified extract (NHEP) was characterized by FTIR and HPLC. In vitro activity determination by chemical method showed that NHEP displayed strong free radical scavenging abilities against the DPPH and ABTS radicals, good reduction power, and hyaluronidase inhibition. The cell viability by CCK-8 assays showed that NHEP had no significant cytotoxicity for B16 and HaCaT cells when the concentration was below 100 μg/mL and 120 μg/mL, respectively. NHEP with a concentration of 20–160 μg/mL can more effectively reduce the ROS level in H_2_O_2_ damaged HaCaT cells compared with 10 μg/mL of VC. The 40 μg/mL of NHEP had similar activity against intracellular melanin production in the B16 melanoma cells compared with 20 μg/mL Kojic acid. Good activities of antioxidation, whitening and protective effect against H_2_O_2_-induced oxidative damage promote the potential for NHEP as a functional raw material in the field of cosmetics and medicine.

## 1. Introduction

*Nymphaea hybrid* is a perennial aquatic herb of *Nymphaea* in *Nymphaeaceae* [1]. It is fragrant, colorful and indispensable as the main plant in features of water purification [2,3]. In the last few decades, scientists have paid more and more attention to *Nymphaea* and found that the extracts of some *Nymphaea* plants displayed good effects of antioxidation [4], whitening [5], antibiosis [6], anti-inflammation [7], liver protection [8], prevention of diabetes and dementia [9,10]. Flavonoids and polyphenols are mainly active substances of *Nymphaea*, which are impressive for the activities of good antioxidation and cell protection [4,11]. Therefore, in-depth study of flavonoids and polyphenols of *Nymphaea* has widespread application prospects in functional food, medicine and cosmetics industries. However, the studies about *Nymphaea*
*hybrid* have been focused on the aromatic components for fragrance application, while there are only a few studies about flavonoids and polyphenols and other active ingredients.

A suitable extraction method is very important for preparing the natural plant active extracts. In recent years, experts have explored new methods of extraction that have a better extraction efficiency, enhanced greenness and shorter extraction time, such as microwave [12], ultrasonic [13], supercritical extraction [14] and enzymatic hydrolysis [15]. Ultrasound-assisted cellulase extraction (UCE) is one of the new methods that has been developed for the extraction of active ingredients from natural plants [16]. The basic principle of enzymatic extraction is to destroy the cell structure using an enzyme as a catalyst under mild reaction conditions so that the active ingredients can be released more quickly [17]. Besides, it has been reported that the enzyme activity can be enhanced by ultrasonic treatment under optimized conditions utilizing its thermal effect, mechanical fluctuant effect and cavitary effect [16]. UCE can not only improve the extraction efficiency but also enhance the quantity of the extracts [18,19,20,21,22,23,24,25,26,27,28]. As we know, there are few studies about extracting flavonoids and polyphenols from the *Nymphaea hybrid*, especially few reports on how to extract flavonoids sufficiently. Hence, in this paper, UCE coupled with the response surface method (RSM) was selected to develop the flavonoids and polyphenols extraction process of *Nymphaea hybrid*.

Natural plant extracts are usually complicated mixtures consisting of a variety of active ingredients. Recent research has shown that flavonoids and polyphenols are the largest groups of secondary metabolites of the *Nymphaea* extracts and are playing an important role in bioactivities [29]. The macroporous resins method is usually used for the further enrichment of active components from the crude extracts [30]. It has the advantages of high adsorption efficiency, simple operation, low cost, reusable and simple equipment [31,32,33,34,35].

Oxidative stress is one of the main risk factors in aging and most age-related diseases. It causes ROS levels to exceed the antioxidant capacity of cells, resulting in oxidative damage [36]. There are a considerable number of chemical [37] and cellular [38] tests for the evaluation of in vitro antioxidant activities of active ingredients for screening natural plant extracts. The chemical tests are easy to operate and have a low cost, which is commonly used for antioxidant and anti-inflammatory assays, such as DPPH, ABTS and other free radical scavenging assays, reduction power and hyaluronidase inhibition assay [39]. The cellular method is more and more prevalent by simulating the environment of the human body for the evaluation of antioxidants. Commonly used cells include fibroblasts, Human keratinocytes, and B16 cells [40]. More and more studies have been carried out into how active substances reduce the UVA/UVB or H_2_O_2_-induced oxidative stress in human cells [41].

The focus of this paper aims to (1) develop and optimize the ultrasound-assisted cellulase extraction process of the flavonoids from *Nymphaea hybrid* flower by RSM coupled with Box–Behnken design; (2) characterize the purified extract (NHEP) by FTIR and HPLC; (3) evaluate in vitro the bioactivities of extract assays to provide theoretical support for the product development and application of *Nymphaea hybrid* extract.

## 2. Results and Discussion

### 2.1. Single-Factor Experiments

The ultrasonic-assisted cellulase method was used to prepare NHE. The process included enzymatical hydrolysis with cellulase followed by ultrasonic extraction of flavonoids and polyphenols. Single-factor experiments and RSM were applied to optimize the preparation process. To explore the influence of different conditions, we developed four factors and five levels of an experimental program in the single-factor experiments. The factors including cellulase amount liquid-to-solid ratio, enzymatic hydrolysis time and ethanol concentration were selected to determine the influence on the content of flavonoids. As shown in Figure 1a, the flavonoid content first increases with the liquid-to-solid ratio from 20:1 (mL/g) to 40:1 (mL/g), then falls in the range of 40:1 (mL/g) to 60:1 (mL/g). The condition for the flavonoid content of NHE reached the maximum with the liquid-to-solid ratio of 40:1 (mL/g). The increasing contact area of solvent and solid is conducive to the diffusion of active substances [27], so the flavonoids content increases with the increase of liquid-to-solid ratio. When the solvent is excessive, the effective concentration of cellulase and substrate concentration non-target active substance dissolved increased to reduce cellulase and substrate reduces, resulting in the flavonoid content of NHE dropping down [22,42]. Figure 1b showed that the flavonoid content increases as the cellulase amount increases from 2% to 4%, but the flavonoid content begins to decline when the addition amount of cellulase is higher than 4%. Cellulase could break down the cell wall into smaller molecules, which is conducive to the release of active substances. Excess cellulase might make a viscous enzyme solution which is not conducive to destroying the cell well and the enzymatic reaction process [43]. As shown in Figure 1c, the flavonoid content reaches a maximum at 60 min of enzymatic hydrolysis time. However, the flavonoid content did not change after 60 min, probably due to the diffusion equilibrium of flavonoids. The concentration of ethanol has an impact on the extraction rate of active components. The concentration of ethanol has an impact on the extraction rate of active components according to the principle of similar miscibility [43]. As shown in Figure 1d, the flavonoid content reaches a maximum when the concentration of ethanol was 50%.

### 2.2. Response Surface Design

Based on the results of the single factor experiments, the liquid-to-solid ratio (A), cellulase amount (B) and ethanol concentration (C) were selected as the independent variables for optimizing the extraction process using the flavonoids content as an index. The results of the Box–Behnken experiment were shown in Table 1. According to different experiments, the flavonoids content of NHE was 20.67–27.06 mg/g. As in Table 2, multiple regression analysis was used to analyze the experimental data. Through the analysis of the multivariate regression fitting, the relationship between the response variables (flavonoids content of NHE, Y) and the quadratic regression equation is shown in the formula as follows:Y = 26.79 + 0.46A + 0.53B + 0.59C − 0.46AB − 0.39AC + 0.41BC − 3.03A^2^ − 1.28B^2^ − 1.67C^2^(1)

The variance analysis of the Box–Behnken regression model results in Table 2 was obtained. The F-value of the equation model is 67.6, and the *p*-value is 0.0001 (*p* < 0.001), indicating that the model has an extremely significant statistical difference, and the model is meaningful. The high correlation coefficient value (R^2^ = 0.9886) indicates that more than 98.86% of the response variability is explained by the model. The *p*-value of lack of fit is 0.2159, the F-value of lack of fit is 2.3293 and the three linear coefficients, three quadratic coefficients and two interaction terms are significant and interact. The results indicate that the quadratic regression model fitted well with the actual situation, and the model could be used to analyze the flavonoid content of NHE.

According to the result of regression analysis, the response surface 3D map and the contour plot maps were drawn in Figure 2. The 3D diagram can intuitively reflect the influence of the interaction of each factor on the response value, and through it, we can find the best parameters and the interactions among the parameters. According to the *p*-value, we can determine two interaction coefficients (AB, BC). Figure 2a shows the interaction between the liquid-to-solid ratio and the cellulase amount (AB) displayed significantly (*p* < 0.05). The flavonoid content increases with the increasing liquid-to-solid ratio and cellulase amount, while at a higher liquid-to-solid ratio (40:1–60:1 mL/g) and cellulase amount (4–6%), the interaction effect was not obvious. When the amount of cellulase reaches a certain point, the flavonoid content may decrease with the increase in cellulase amount and enlarge the liquid-to-solid ratio. The reason for this may be due to the appropriate cellulase amount and an enlargement of the liquid-to-solid ratio will improve the solubility and increase the mass transfer rate, but the excessive cellulase amount with too high a liquid-to-solid ratio will affect the reaction efficiency and mass transfer efficiency, resulting in a flavonoid content decrease with the liquid-to-solid ratio. A significant interaction effect (*p* < 0.05) between cellulase amount and ethanol concentration (BC) was also observed in Figure 2c.

According to the response surface experiment, the optimal process conditions for extraction were as follows: the liquid-to-solid ratio was 40.45:1 (mL/g), cellulase amount was 5.23% (*w*/*w*), the ethanol concentration was 52% (*v*/*v*). Under such conditions, the predicted flavonoid content of NHE was 26.92 mg/g. To verify these optimal conditions, the experiment was repeated five times and the average actual flavonoid content of NHE was 27.26 ± 0.64 mg/g (*n* = 5), which was close to the predicted value and proved the validity of the models.

### 2.3. AB-8 Macroporous Resin

#### 2.3.1. Static Adsorption and Desorption

The static adsorption and desorption curves of AB-8 macroporous resin for the flavonoid content of NHE were shown in Figure 3. As shown in Figure 3, the adsorption capacity of resin AB-8 is greater at 0–80 min; the adsorption capacity is slowly increasing at 80–240 min; and the adsorption capacity is not changed significantly after 240 min. This suggests that the adsorption equilibrium occurred and the adsorption capacity was 16.35 ± 0.25 mg/g (*n* = 3). The flavonoids into the resin AB-8 were desorbed, and the desorption ratio increased rapidly first and then slowed down within the first 240 min. After 240 min, the flavonoid content in the solution of desorption did not change significantly when the desorption capacity was 15.61 ± 0.25 mg/g (*n* = 3).

The effects were determined with pH (Figure 3b), sample concentration (Figure 3c), ethanol concentration (Figure 3d) on the static adsorption and desorption experiments. The solutions of NHE were adjusted to a different pH value (2.0, 3.0, 4.0, 5.0 and 6.0) by using 2.0 mol/L HCl and 2.0 mol/L NaOH solutions. Because flavonoids have a phenolic hydroxyl structure and are acidic, they can exist in a molecular state and can be absorbed into the resin by van der Waals force under the condition of the weak acid [44]. Therefore, the flavonoid content was highest at pH = 3. The solutions of NHE were diluted with distilled water to different concentrations (0.33, 0.49, 0.65, 0.82, 0.98 and 1.15 mg/mL). With the increase in concentration of the sample solution, the flavonoid adsorption capacity of resin AB-8 increased first and then remained stable. The high concentration of the sample solution blocked resin adsorption due to the interaction of the hydroxyl of flavonoids with water. The adsorption capacity of the high solution decreased due to the increase in impurities in the solution [33,45].

Ethanol is commonly used as an eluent. It has a good solubility for flavonoids. A small amount of water solution can dissolve the part of water-soluble components and improve the desorption of flavonoids [33,46]. The absorbed flavonoids into the resin AB-8 were desorbed with ethanol of different concentrations (50%, 60%, 70%, 80%, 90%). As shown in Figure 3d, with the increase of ethanol concentration, the flavonoid content of the solution was increased first and then decreased. The 70% (*v*/*v*) was the optimum ethanol concentration.

#### 2.3.2. Dynamic Adsorption and Desorption

The influence of sample volume (a) and adsorption flow velocity (b) were determined on the dynamic adsorption capacity of flavonoids into the resin AB-8 in Figure 4. The abscissa was the sample volume, and the ordinate was the flavonoid concentration of the sample solution after adsorption. As shown in Figure 4a, with the increase of sample volume, the adsorption capacity of AB-8 decreased in the range of 0–70 mL and then did not significantly change in the range of 70–120 mL. The adsorption capacity of resin AB-8 was limited, while the appropriate adsorption volume can effectively improve [33]. Figure 4b shows that the samples were into the resin AB-8 with the flow rate of 1.2–7.2 BV/h, in which the adsorption capacity of AB-8 decreased on the NHE flavonoids. When the flow rate was too high, some active components were not adsorbed by the resin AB-8 and leaked out [47]. The changing trend of flavonoid concentration was similar in the effluent at 1.2 BV/h and 2.4 BV/h. Therefore, the flow rate of 2.4 BV/h was selected for adsorption with higher efficiency.

The dynamic adsorption (c) and desorption (d) curves were drawn in Figure 4. The concentration of flavonoids of effluent was increased gradually at the flow rate of 2.4 BV/h. When the sample volume was 70 mL in Figure 4c, the flavonoid capacity in the effluent solution did not change significantly. As shown in Figure 4d, the flavonoids into the resin AB-8 were firstly increased and then decreased with the eluent of 70% ethanol. When the flavonoids concentration of eluent resolution approached zero, the adsorbed flavonoids of resin AB-8 were completely eluted.

NHEP was prepared by the purification of NHE with AB-8 macroporous resin. As shown in Table 3, the content of flavonoids and polyphenols in the NHEP sample was higher than in the NHE sample with the same dry weight. After purification, the contents of flavonoids and polyphenols in NHEP increased from 55.60 mg/g to 157.39 mg/g and from 414.97 mg/g to 888.63 mg/g, respectively. This result showed that it was efficient for the enrichment of flavonoids and polyphenols of NHE by AB-8 macroporous resin.

### 2.4. Activity Determination by Chemical Method

To explore in vitro antioxidant activity of NHE and NHEP, the free radical scavenging abilities against ABTS, DPPH, and reduction power, were detected compared with VC. As shown in Figure 5, the IC_50_ values of the DPPH radical of NHE, NHEP and VC were 11.09 ± 0.15 μg/mL, 3.28 ± 0.03 μg/mL and 5.18 ± 0.06 μg/mL, respectively. The IC_50_ values of the ABTS radical of NHE, NHEP and VC were 18.21 ± 0.05 μg/mL, 10.50 ± 0.10 μg/mL and 7.77 ± 0.04 μg/mL, respectively. The reduction power of NHE, NHEP, VC for Fe^3+^ were strong with high absorbances [48]. The absorbances were 2.20 ± 0.04, 2.36 ± 0.04 and 2.60 ± 0.07 in the 1 mg/mL concentration, respectively. This result indicated that the scavenging free radical abilities and reduction power of NHEP were similar to VC and stronger than that of NHE. NHE and NHEP were also evaluated for the anti-inflammatory activity by using hyaluronidase inhibition assays. The IC_50_ value of NHE and NHEP in hyaluronidase inhibition assay was 0.55 ± 0.01 mg/mL and 0.32 ± 0.01 mg/mL. The inhibition rate of 10 μg/mL dipotassium glycyrrhizinate (Dg) was 94.23 ± 1.72%. This result showed that both NHE and NHEP display an anti-inflammatory effect, and NHEP has a stronger anti-inflammatory effect than NHE. Hence, in vitro bioactivity evaluation by chemical assays showed that both NHE and NHEP displayed strong free radical scavenging abilities against the DPPH and ABTS radicals, good reduction power, and anti-inflammatory effect.

### 2.5. Characterization of FTIR and HPLC

Infrared spectroscopy analysis can be used as a powerful tool to characterize and identify functional groups present in the compounds. As shown in Figure 6, the absorption peak at the vicinity of 3424 cm^−1^ is relatively wide and strong, which is attributed to the stretching vibration of O−H. The signal in wavelength of 2930 cm^−1^ is assigned to the absorption peak of stretching vibration of the C−H bond of aliphatic hydrocarbon [49]. The band at 3000–3100 cm^−1^ is due to the stretching vibration of C−H of the aromatic ring. The aromatic C−H bond was out-of-plane bending vibration, resulting in absorption peaks at 834 and 760 cm^−1^ wavelengths [50]. A characteristic absorption peak at 1718 cm^−1^ may be assigned to the carboxylic acid of a sharp stretching band of the C=O. The peaks at 1613 and 1449 cm^−1^ wavelength belong to the C=C bond of the aromatic ring vibrates [51]. The absorption peaks at 1512 cm^−1^ may be assigned to the aromatic skeletal vibrations, ring breathing with C−O starching vibration [50]. The adsorption peaks at 1449 and 1349 cm^−1^ are assigned to the bending vibrations of −CH_3_ and −CH_2_. The peaks at 1210 and 1038 cm^−1^ wavelength belong to the stretching vibration of C−H and C−O bonds [52]. The characteristic peaks of the benzene ring, C−O stretching vibration and stretching vibration of O−H, indicate the presence of phenolic hydroxy groups [53].

The total polyphenol and flavonoid compositions of NHEP were identified by the characterization of HPLC in Figure 7 and Table 4. This result showed some of the main NHEP bioactive phytochemicals, including gallic acid (1.60 ± 0.36 mg/g), corilagin (65.60 ± 3.36 mg/g), ellagic acid (97.72 ± 0.92 mg/g), rutin (34.41 ± 4.24 mg/g), myricetin (3.13 ± 0.87 mg/g), quercetin (4.64 ± 0.15 mg/g), naringin (4.03 ± 0.24 mg/g) etc. The contents of corilagin, ellagic acid and rutin in NHEP were higher among these components. These standard substances have been reported to have excellent antioxidant activity, which further indicated that NHEP has great potential as a natural antioxidant. Uddin et al. identified 9 bioactive phytochemicals from methanolic extract of *Nymphaea nouchali* using an HPLC system [54]. Among these compounds, rutin (39.44 mg/g), myricetin (30.77 mg/g), ellagic acid (11.05 mg/g), gallic acid (5.33 mg/g) and quercetin (0.90 mg/g) were polyphenols or flavonoids, which were also identified as bioactive ingredients of NHEP in this work. We found the ellagic acid content (97.72 ± 0.92 mg/g) of NHEP was much greater than that (11.05 mg/g) of *Nymphaea nouchali* and that (6.16 ± 0.019 mg/g) of ethanolic extract of Water Lily (*Nymphaea Tetragona Georgi*) [55].

### 2.6. HaCaT Cells Experiments

#### 2.6.1. Effects of NHEP and H_2_O_2_ on Cell Viability of HaCaT Cells

CCK-8 kit assay was used to determine the cytotoxicity of NHEP and H_2_O_2_ on HaCaT cells. As shown in Figure 8a, HaCaT cells were cultured with NHEP in the concentration range of 10–120 μg/mL for 24 h, and the cell viability was higher than 90% of the control group, indicating that there was no cytotoxicity. The higher the dose of NHEP, the lower the cell viability observed. When the concentration of NHEP rose to 160 μg/mL and 200 μg/mL, the cell viability decreased to 88.79% ± 2.64% (*p* < 0.001) and 81.14% ± 3.66% (*p* < 0.001). As shown in Figure 8b, the culture time of the cells treated with NHEP in the same concentration range was extended from 24 h to 48 h. The cell viability was higher than 90% of the control group in the NHEP concentration range of 10–80 μg/mL. The cell viability decreased gradually when the concentration of NHEP increased to 120 μg/mL (88.17% ± 5.16%, *p* < 0.001), 160 μg/mL (80.75% ± 4.87%, *p* < 0.001) and 200 μg/mL (72.85% ± 3.58%, *p* < 0.001). We can obviously find that while the culture time of the cells extended, the cytotoxicity significantly increased as the cell viability decreased by 10.81%, 8.04%, 8.29%, respectively when the cells were treated with a higher concentration of NHEP (120–200 μg/mL). The CCK-8 assay results were consistent with cell growth density and cellular morphology, as shown in Figure 8c. Therefore, with the concentration of NHEP being higher and the incubation time of HaCaT being longer, the cytotoxicity of cells was stronger, affecting cell growth activity and resulting in cell shedding and death [56].

H_2_O_2_ can be used as an inducer to promote intracellular ROS production in certain concentrations [57]. In order to confirm an appropriate concentration for the H_2_O_2_-induced injury model in vitro, the nominal concentration of H_2_O_2_ (25–300 μM) exposed to HaCaT cells was assessed by the CCK-8 kit. As shown in Figure 9a, the cell viability of HaCaT cells decreased gradually to 50% (*p* < 0.001) with a concentration of H_2_O_2_ increased to 100 μM. Therefore, 100 μM of H_2_O_2_ was selected to induce cellular ROS oxidation damage on HaCaT cells in the subsequent experiments.

As shown in Figure 9b, HaCaT cells were cultured first with 10–200 μg/mL NHEP and then treated with 100 μM H_2_O_2_ for 2 h. Compared with the control group treated only with H_2_O_2_, HaCaT cells had higher cell viability when pretreated with NHEP in the range of 20–160 μg/mL. The higher concentration of NHEP, the higher the HaCaT cell viability. The cell viabilities of HaCaT cells cultured with 20, 40, 80, 120, 160 μg/mL NHEP were 52.66% ± 2.36% (*p* < 0.05), 60.19% ± 3.71% (*p* < 0.01), 66.85% ± 2.00% (*p* < 0.001), 75.77% ± 3.79% (*p* < 0.001) and 82.17% ± 4.70% (*p* < 0.001), respectively. Although 160 μg/mL NHEP had a low cytotoxicity compared with HaCaT cells, it displayed a better inhibition against the stimulation of H_2_O_2_. The cell viabilities of 80–160 μg/mL NHEP pretreatment groups were higher than that of the 10 μg/mL VC pretreatment group (64.44% ± 3.00%, *p* < 0.001). It was reported that polyphenols and flavonoids can effectively inhibit the generation of intracellular ROS and protect cells against oxidative damage [41]. In this case, the content of polyphenols and flavonoids increased with the increase of NHEP concentration. The polyphenols and flavonoids of NHEP may positively contribute to the protection of HaCaT cells against oxidative damage induced by H_2_O_2_.

#### 2.6.2. Effects of NHEP on the ROS Level of HaCaT Cells with Stimulation of H_2_O_2_

Excessive ROS may destroy the structural integrity of the cell and eventually lead to oxidative damage of organisms [57]. H_2_O_2_ is a precursor of many radicals and can increase the intracellular ROS levels by crossing the cell membrane [58,59]. In this study, the HaCaT cells were first induced to produce ROS with 100 μM H_2_O_2_ and then treated with NHEP of different concentrations.

The effect of intracellular ROS level in the HaCaT cells was detected by a fluorescent probe DCFH-DA. The percentage of ROS proportion was taken in the control group as the index. As shown in Figure 10, the ROS levels and fluorescent intensity in HaCaT cells significantly increased after the stimulation of H_2_O_2_. The ROS levers of the stimulated injury group were 295.17% ± 9.29% (*p* < 0.001), which is significantly higher than that of the control group and NHEP post-treatment groups. With the increasing concentration of NHEP, the ROS levels reduced gradually, indicating that NHEP in the concentration range of 10–160 μg/mL can efficiently inhibit the ROS production in HaCaT cells stimulated by H_2_O_2_. The ROS levers of 20–160 μg/mL NHEP groups were lower than that of the VC group (220.17% ± 9.12%, *p* < 0.01), and displayed weaker fluorescence intensity. The ROS production of 20–160 μg/mL NHEP were 202.35% ± 10.65% (*p* < 0.01), 169.74% ± 7.88% (*p* < 0.01), 132.43% ± 11.25% (*p* < 0.001), 116.87% ± 4.22% (*p* < 0.001) and 108.52% ± 5.75% (*p* < 0.001), respectively. In conclusion, NHEP has a better protection ability antioxidative damage. NHEP can effectively reduce the ROS levels in HaCaT cells induced by H_2_O_2_ so that NHEP has good cell protection against oxidative damage in the range of 20–120 μg/mL.

### 2.7. B16 Cell Experiments

#### 2.7.1. Activity of B16 Cells

The cytotoxicity of 10–140 μg/mL NHEP on B16 cells was determined by the CCK-8 kits. As shown in Figure 11a, B16 cells were cultured with 10–140 μg/mL NHEP for 24 h. The result showed that it was no significant cytotoxicity with the cell higher than 90% in the NHEP concentration range of 10–120 μg/mL. The cell viability decreased to 84.66% ± 3.45% (*p* < 0.001) when the NHEP concentration was 140 μg/mL. As shown in Figure 11b, B16 cells were cultured with 10–140 μg/mL NHEP for 48 h, and results showed that it was not significant with the cell viability higher than 90% in the NHEP concentration range of 10–100 μg/mL. The cell viabilities of 100, 120 and 140 μg/mL decreased to 89.07% ± 3.17% (*p* < 0.001), 72.98% ± 2.96% (*p* < 0.001) and 63.54% ± 3.73% (*p* < 0.001), respectively. Comparing the results of different culture times, we can obviously find that the culture time of the cells extended the cytotoxicity increased significantly with cell viability decreased by 10.79%, 18.26% and 21.11%, respectively, when the cells were treated with a higher concentration of NHEP (100–140 μg/mL). It was reported that flavonoids and polyphenols would affect anti-tumor cell proliferation and apoptosis [60,61]. In conclusion, with the increase in concentration and incubation time, the cytotoxicity of B16 cells was significantly enhanced. As shown in Figure 11c, the cell growth density decreased at high concentrations. The results were consistent with the data reported by Teng et al. that a high concentration of *Nymphaea* extract may induce apoptosis of B16 cells [62,63].

#### 2.7.2. Effects of NHEP on B16 Melanin Production

As shown in Figure 12, 10–140 μg/mL NHEP effectively inhibited the melanin production on B16 cells, and the higher concentration of NHEP and the stronger the inhibition ability. This result indicated that NHEP has a better effect on whitening by inhibiting melanin production. When B16 cells were cultured with NHEP in the range of 10–80 μg/mL for 24 h or 48 h and no significant cytotoxicity. The melanin content of B16 cells was 67.35% ± 2.36% (*p* < 0.001) when treated with 40 μg/mL NHEP. A similar melanin content of B16 cells (68.57% ± 4.47%, *p* < 0.001) was produced in contrast with the positive control group (20 μg/mL KA). When the NHEP concentration increased to 100–140 μg/mL, the melanin content obviously decreased to 42.65% ± 2.34% (*p* < 0.001), 36.53% ± 1.81% (*p* < 0.001) and 24.69% ± 5.18% (*p* < 0.001), respectively. This may be due to the fact that NHEP can induce apoptosis of B16 cells or inhibit tyrosinase activity [62,63].

## 3. Materials and Methods

### 3.1. Materials

*Nymphaea hybrid* flowers were purchased from the Liandao Agricultural Development Co., Ltd. (Shanghai, China). *Nymphaea hybrid* flowers were rinsed with distilled water and frozen. Then, the frozen *Nymphaea hybrid* flowers were placed into a vacuum freeze-dryer (TF-FD-1, Tuoyuan Co., Ltd., Shanghai, China) for freeze-drying to a constant weight. Then *Nymphaea hybrid* flowers were crushed into a fine powder (50 mesh) by a shredder and stored in air-sealed bags at a low temperature (−20 °C) for use. The AB-8 macroporous resins were purchased from the San Xing Resin Co., Ltd. (Anhui, Bengbu, China).

### 3.2. Chemical Reagents

Ultrapure water, anhydrous ethanol, Folin-phenol, AlCl_3_, Na_2_CO_3_, HCl, NaOH, CaCl_3_, NaH_2_PO_4_, Na_2_HPO_4_, KBr, acetic acid, sodium acetate, acetylacetone, 4-Dimethylaminobenzaldehyde, DPPH, ABTS, Dipotassium glycyrrhizinate, cellulase, potassium persulfate, hyaluronidase (Sigma, H3506, Shanghai, China), sodium hyaluronate (Sigma, H5388, Shanghai, China), DMEM, CCK-8 assay, PBS, pancreatin, reactive oxygen species assay kit, H_2_O_2_, methanol, formic acid, acetonitrile and other chemicals, rutin standard, gallic acid standard, ellagic acid standard, corilagin standard, myricetin standard, quercetin standard, naringin standard, all reagents is analytical grade. All reagents, solvents and standards were purchased from MACKLIN (Shanghai, China). HaCaT cells and B16 mouse melanoma cells were purchased from the Shanghai Cell Bank of the Chinese Academic of Sciences.

### 3.3. Preparation of NHE

NHE was prepared by a method of UCE [42]. Degrease *Nymphaea hybrid* flowers powder with petroleum ether and weighs 1.0 g. Then *Nymphaea hybrid* flowers powder was mixed with PBS buffer solution (20:1–60:1, mL/g), and adjusted the pH of the solution to be five. After adding an amount of cellulase (1–5%, *w*/*w*), the mixture was enzymatically hydrolyzed for 20–100 min at the thermostatic water bath of 50 °C. Then, absolute ethanol was added to the mixture (20–60% ethanol concentration, *v*/*v*) and the mixture was associated at the ultrasonic power of 150 W and the temperature of 50 °C for 30 min. Finally, the resulting mixture was centrifuged at 10,000 rpm for 15 min. The liquid NHE was separated from the supernatant.

### 3.4. Flavonoids and Polyphenol Determination

In this study, the total flavonoids were determined using the colorimetric AlCl_3_ method [4]. The total flavonoids content was expressed in the dry weight of rutin equivalent and measured absorbance at 415 nm using an ultraviolet spectrophotometer (ALPHA-1860, Puyuan, Shanghai, China). The horizontal coordinate was the concentration of rutin standard, and the vertical coordinate was absorbance to make a five-point calibration line (y = 0.0078x − 0.0008, R^2^ = 0.9992). The total flavonoids of NHE were detected by the above method.

The total polyphenols were determined using the method of Folin–Ciocalteu [64]. It was expressed in the dry weight of gallic acid equivalent and measured absorbance at 760 nm using an ultraviolet spectrophotometer. The horizontal coordinate was the concentration of gallic acid standard, and the vertical coordinate was absorbance to make a five-point calibration line (y = 0.0431x + 0.0089, R^2^ = 0.9991). The total polyphenols of NHE were detected by the above method.

### 3.5. Response Surface Experiment

Based on single-factor results, Design-Expert software 7.1.3 was used for the experimental design, the result analysis and response prediction. According to the Box–Behnken experiment, the extraction process was optimized by three factors and three levels. The flavonoid content was taken as the index.

### 3.6. AB-8 Macroporous Resin Experiment

NHE was prepared by the experiment method of the Section 2.3. As an alcohol extract, NHE consisted of a large number of total flavonoids and polyphenols. NHE was purified to obtain NHEP by AB-8 macroporous resin [32,33]. In this experiment, AB-8 macroporous resins are weak polar resins, milk-white spherical particles, and the specific surface area is 480–520 m^2^/g. Macroporous resin can be selectively absorbed flavonoids and polyphenols constituents by electrostatic force, hydrogen bonding interactions, complexation and size sieving [45]. Enrichment of these active substances, the flavonoid content of NHE was taken as the index. Different influencing factors were explored to optimize the experiment. Such as AB-8 macroporous resin static/dynamic adsorption and desorption experiments. The influence was tested for adsorption of concentration of NHE (0.33–1.15 mg/mL), pH (2–6) and desorption of ethanol concentration (50–90%) on the static adsorption and desorption experiments, and sample volume (120 mL) and sample flow rate (1.2–7.2 BV/h) on the dynamic experiments. Dynamic experiments: Chromatographic columns: diameter × length (22 × 300 mm); the quality of the resin was 5 g; the volume of the resin bed (BV) was 25 mL. Powders of NHE and NHEP, weighing 10.0 mg, was determined the content of polyphenols and flavonoids to test the purification effect of AB-8 macroporous resins. These are examples 2 and 3 of an equation:(2)$$Q=\frac{C_{0}-C_{r}\times V_{0}}{m}$$
(3)$$P=\frac{C_{d}\times V_{1}}{m}$$
where Q is the absorption capacity of the resin (mg/g), P is the desorption capacity of the resin (mg/g), C_0_ and C_r_ are flavonoids content of NHE at the beginning and at time t (mg/mL), V_0_ is the volume of sample (mL), m is the weight of the resin (g), C_d_ is flavonoids content of the desorption solution (mg/mL), V_1_ is the desorption solution volume (mL).

### 3.7. Antioxidant and Anti-Inflammatory Activity Analysis

In this study, the raw extract NHE and its purified product NHEP were lyophilized into powder. Their DPPH, ABTS radical scavenging activities and reducing power were measured by an ultraviolet spectrophotometer using VC as the positive control according to the method of Kunnaja et al. [37] and Shaddel et al. [65], respectively. According to the experiment method of Mancarz et al. [66], the anti-inflammatory activity of NHE and NHEP was determined by the inhibition of hyaluronidase activity using dipotassium glycyrrhizinate (Dg) as the positive control.

### 3.8. Characterization of NHEP by HPLC and FTIR

In this study, the HPLC experiment of the instrument used the Agilent 1260 high-performance liquid chromatography system (Agilent Technologies Inc, Santa Clara, CA, USA). Gallic acid, ellagic acid, rutin, corilagin, myricetin, quercetin, naringin standards and NHEP were determined. The powder of NHEP was prepared to the solution of 1 mg/mL. Chromatographic conditions [29,54,55]: C18 column (4.6 mm × 250 mm, 5 μm); mobile phase: 0.1% formic acid in water (A) and acetonitrile (B); gradient elution conditions were 5–13% B for 0–10 min, 13% B for 10–20 min, 13–20% B for 20–35 min, 20–45% B for 35–55 min, 45–100% B for 55–60 min; sample injection volume was 10 μL; flow rate: 1 mL/min; column temperature: 35 °C; detection wavelength: 280 nm.

The powder of NHEP was mixed with spectroscopic KBr, pressed into a sheet for IR analysis using the FTIR spectrometer (Shimadzu Corporation, Shanghai, China) in the range of 4000–500 cm^−1^ [67].

### 3.9. HaCaT Cell Experiments

The experiment referred to the method of ZHANG J et al. [57,68], the activity of HaCaT cells was determined by the CCK-8 assay. The HaCaT cells of lines growth state were chosen and cultured in DMEM medium containing 10% fetal bovine serum by the culture flask of the adherent cell of 25 cm^2^. The cells were seeded in sterile 96-well plates according to the density of 10,000 per well for 24 h and 48 h into the carbon dioxide incubator at 37 °C and 5% CO_2_. The NHEP solution of 1 mg/mL was prepared to remove bacteria and impurities by a 220 nm microporous membrane and the sample was prepared into a mixed medium with a concentration of 10–200 μg/mL. The activity of HaCaT cells was determined by the CCK-8 assay after incubation for 4 h. The optical density (OD) value was measured at the wavelength of 450 nm by the enzyme reader (TECAN, Infinite M200 Pro, Shanghai, China), which was expressed as the percentage of HaCaT cell activity in the control group. Then, the HaCaT cells were seeded in sterile 6-well plates according to the density of 200,000 per well for 24 h and 48 h. The morphology of cells was observed by microscope (Axio Vert.A1, Shanghai, China).

H_2_O_2_ is one of the ways that induced intracellular production of ROS [69,70]. The HaCaT cells were treated with a nominal concentration of H_2_O_2_ (25–300 μm) for 2 h. Then, the activity of cells was detected by the CCK-8 assay.

In this experiment, the level of reactive oxygen species (ROS) was detected by a fluorescent probe DCFH-DA in the HaCaT cells [71]. The cells were seeded in black sterile 96-well plates according to the density of 10,000 per well. After 24 h of culture in the mixed medium of the NHEP of 10–160 μg/mL, HaCaT cells were stimulated with 100 μM H_2_O_2_ for 2 h. Then, the cells were mixed with the DCFH-DA fluorescent probe for 30 min. The fluorescence intensity was detected at the excitation wavelength of 488 nm and the emission wavelength of 525 nm, which was expressed as the percentage of ROS in the control group of HaCaT cells.

### 3.10. B16 Mouse Melanoma Cells Experiments

The experiment referred to the method of AIMVIJARN P et al. [63], the activity of B16 cells was determined by the CCK-8 assay. The cells were seeded in sterile 96-well plates according to the density of 6000 per well for 24 h and 48 h. The samples were prepared into a mixed medium with a concentration of 10–140 μg/mL. Then, the B16 cells were seeded in sterile 6-well plates according to the density of 100,000 per well for 24 h and 48 h. The morphology of cells was observed by microscope.

The melanin content of B16 cells was detected by referring to the method of Teng et al. [62] The sample was mixed medium with a concentration of 10–80 μg/mL. The B16 cells were seeded in sterile 6-well plates according to the density of 100,000 per well for 72 h. Then, the cells were separated by trypsinization and disrupted by using 10% DMSO with 1 M NaOH at 80 °C for 1 h. The melanin content was determined at 405 nm, which was expressed as the percentage of melanin content in the control group of B16 cells.

### 3.11. Statistical Analysis

Design-Expert software 7.1.3 was used for response surface analyses. Other Statistical analyses were performed with OriginPro 2021b. All experiments tested replicate at least three times. The significant represented that sample group compared with the control group, *p* < 0.05, *p* < 0.01 and *p* < 0.001 were performed *, ** and ***, respectively. The data shows the mean ± S.D.

## 4. Conclusions

In this paper, ultrasound-assisted cellulase extraction method associated with macroporous resin purification step was successfully applied to enrich antioxidant flavonoids and polyphenols from *Nymphaea hybrid* flowers for the first time. The optimum extraction conditions obtained by RSM were as follows: liquid-to-solid ratio of 40.45:1 (mL/g), cellulase ratio of 4.23%, and ethanol concentration of 52%. After purification by macroporous resins, the flavonoid content increased from 55.60 mg/g (NHE) to 157.37 mg/g (NHEP), and the polyphenols content increased from 414.97 mg/g (NHE) to 888.63 mg/g (NHEP), respectively. Seven major flavonoids or polyphenols in NHEP were identified and quantified by the HPLC method. The chemical evaluation method of the antioxidant and anti-inflammatory effect of NHEP was determined by scavenging free radicals (DPPH, ABTS), reduction power and inhibiting hyaluronidase assay. The intracellular antioxidant and whitening effects of NHEP on HaCaT cells and B16 cells were detected. These experiments results indicated that the NHEP displayed significantly antioxidant, whitening, anti-inflammatory effects and the protective effect against H_2_O_2_-induced oxidative damage in human cells, which promote the potential as a functional raw material in the field of cosmetics and medicine.

## Figures and Tables

**Figure 1 molecules-27-01914-f001:**
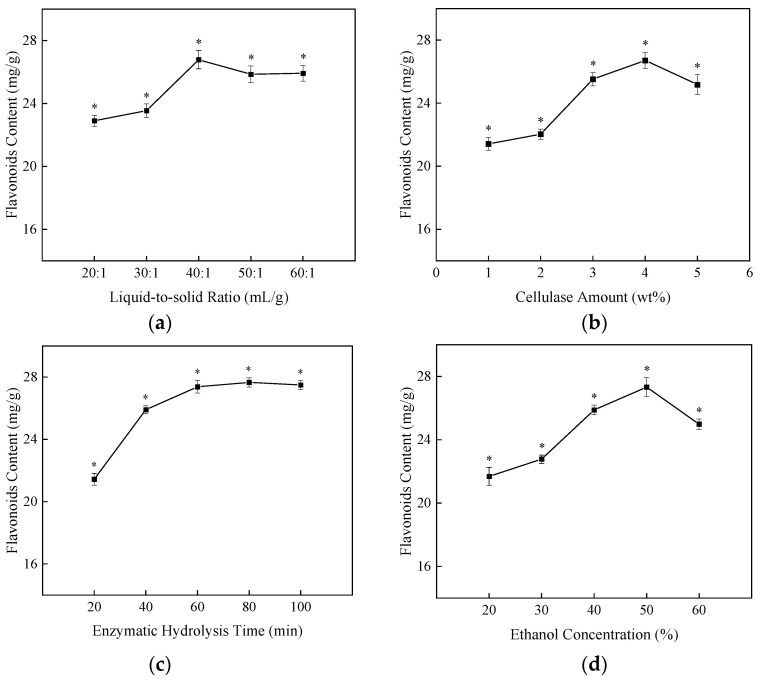
The influence of different factors on the content of flavonoids in NHE. (**a**) Liquid-to-solid ratio, (**b**) Cellulase amount (wt%), (**c**) Enzymatic hydrolysis time (min), (**d**) Ethanol concentration (%). The data shows as the mean ± S.D. (*n* = 3). *: *p* < 0.05 compared between groups.

**Figure 2 molecules-27-01914-f002:**
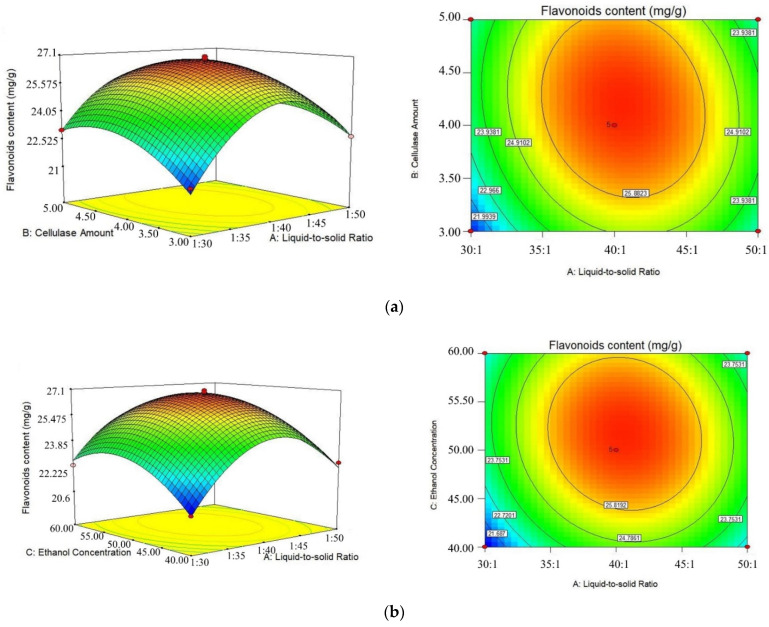
Response surface 3D and contour plots of different factors to effect the flavonoid content of NHE. (**a**) Liquid-to-solid ratio and cellulase amount; (**b**) Liquid-to-solid ratio and ethanol concentration; (**c**) Cellulase amount and ethanol concentration.

**Figure 3 molecules-27-01914-f003:**
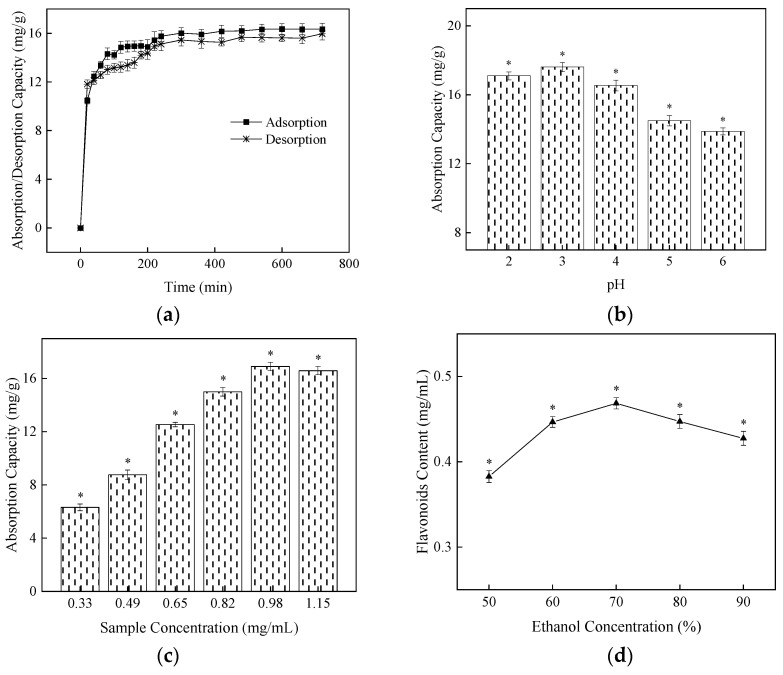
Effects of different factors on static adsorption and desorption of AB-8. (**a**) Adsorption and desorption curve; (**b**) Influence of pH on adsorption; (**c**) Influence of sample concentration on adsorption; (**d**) Influence of ethanol concentration on desorption. The data shows as the mean ± S.D. (*n* = 3). *: *p* < 0.05 compared between groups.

**Figure 4 molecules-27-01914-f004:**
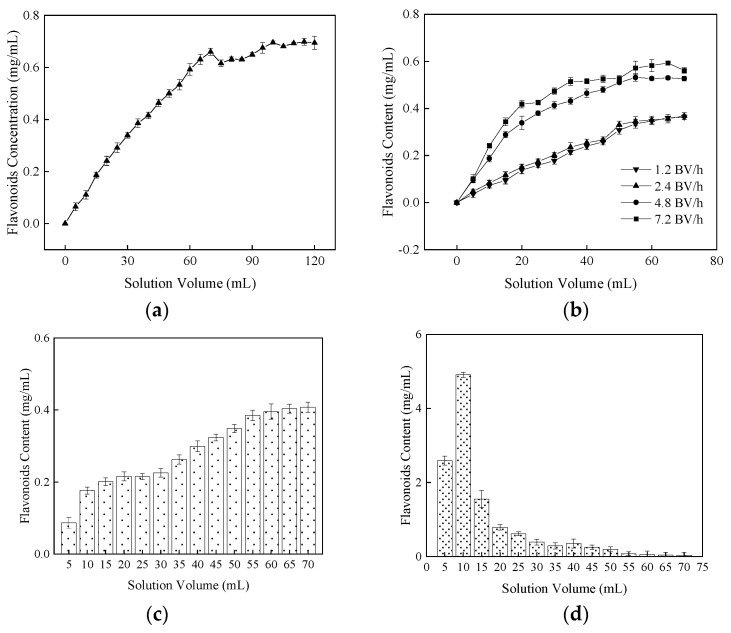
Effects of different factors on dynamic adsorption and desorption of AB-8. (**a**) Sample volume, (**b**) Adsorption flow velocity, (**c**) Adsorption and (**d**) Desorption curve. The data are shown as the mean ± S.D. (*n* = 3). The volume of the resin bed (BV) is 25 mL.

**Figure 5 molecules-27-01914-f005:**
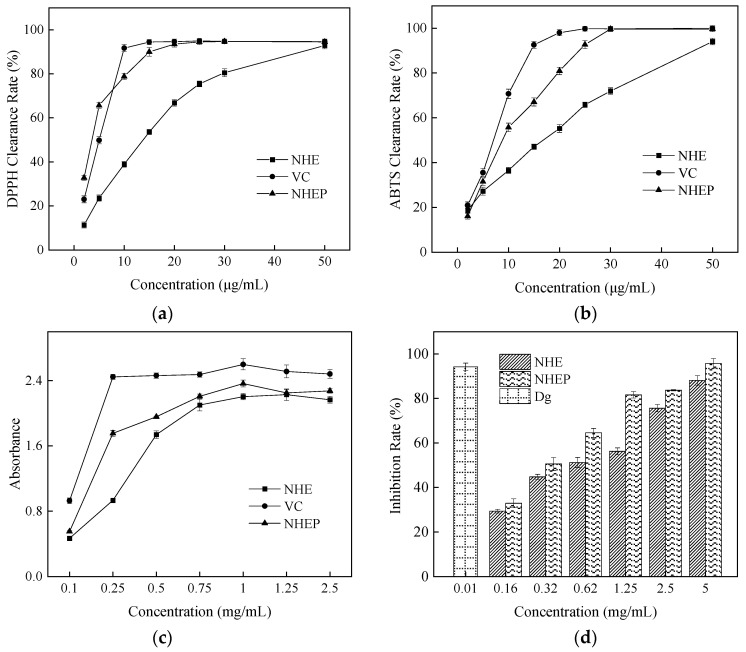
Activities of the NHE, NHEP and positive control by chemical method. (**a**) DPPH, (**b**) ABTS, (**c**) Reduction power, (**d**) Hyaluronidase inhibition assay. Dg stands for Dipotassium glycyrrhizinate, and the concentration is 10 μg/mL. The date is shown as the mean ± S.D. (*n* = 3).

**Figure 6 molecules-27-01914-f006:**
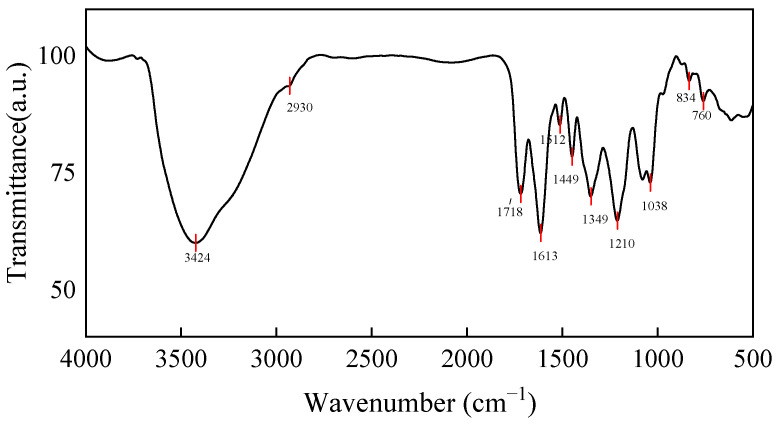
FTIR-NHEP illustrating structural features.

**Figure 7 molecules-27-01914-f007:**
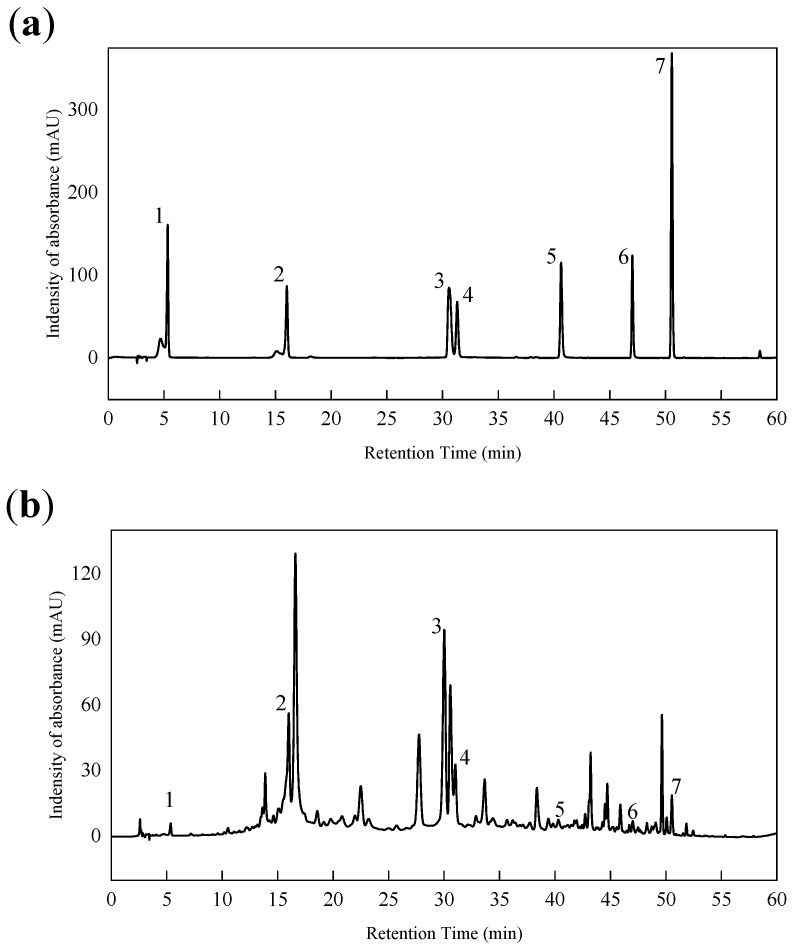
HPLC of mixed standards (**a**) and NHEP (**b**). 1-gallic acid, 2-corilagin, 3-ellagic acid, 4-rutin, 5-myricetin, 6-quercetin, 7-naringin.

**Figure 8 molecules-27-01914-f008:**
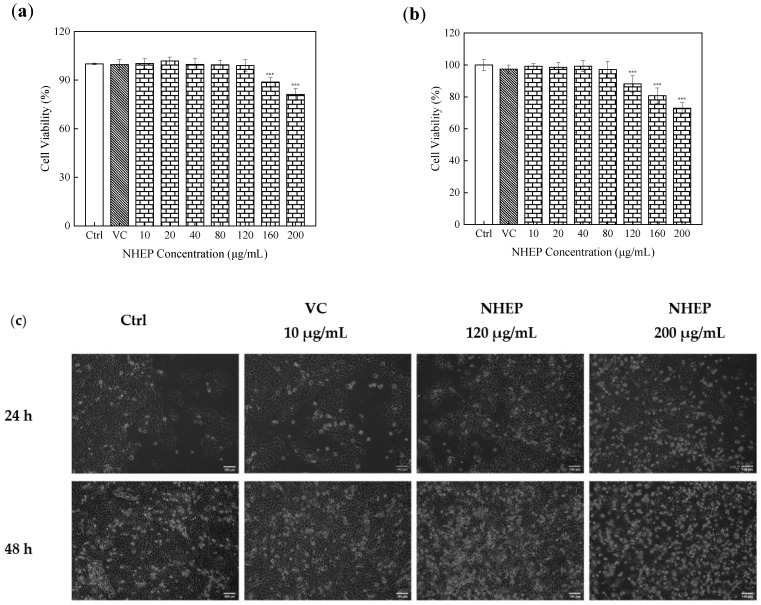
Cytotoxicity of NHEP on HaCaT cells. (**a**) cell viability of the HaCaT cells cultured with the NHEP for 24 h, (**b**) cell viability of the HaCaT cells cultured with the NHEP for 48 h, (**c**) cellular morphology of the HaCaT cells cultured with the NHEP for 24 h and 48 h (scale bar is 100 μm). VC stands for Vitamin C, and the concentration is 10 μg/mL. Significance: compared with the control group, *** *p* < 0.001. The data shows as the mean ± S.D. (*n* = 5).

**Figure 9 molecules-27-01914-f009:**
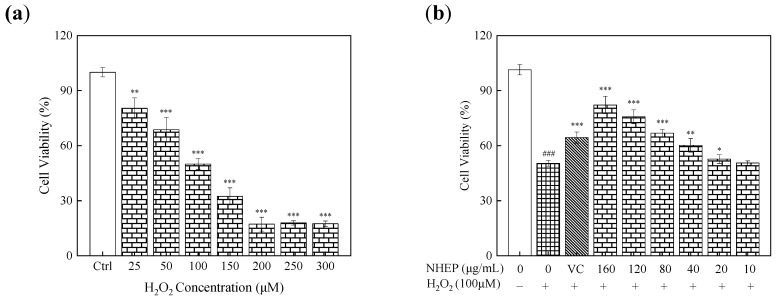
Cytotoxicity of H_2_O_2_ and NHEP on HaCaT cells. (**a**) viability of the HaCaT cells cultured with the H_2_O_2_, (**b**) viability of the HaCaT cells cultured with H_2_O_2_ and NHEP. VC stands for Vitamin C, and the concentration is 10 μg/mL. Significance: ### *p* < 0.001 compared with the control group. * *p* < 0.05, ** *p* < 0.01, *** *p* < 0.001 compared with the H_2_O_2_ stimulation group. The data shows as the mean ± S.D. (*n* = 5).

**Figure 10 molecules-27-01914-f010:**
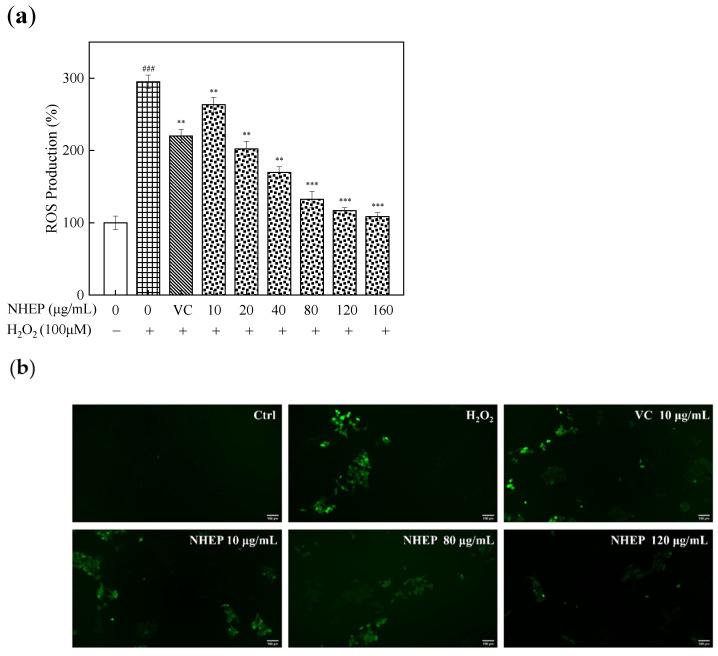
Effects of NHEP on the ROS level of HaCaT cells with stimulation of H_2_O_2_: (**a**) ROS production, (**b**) fluorescence intensity by detected the DCFH-DA. VC stands for Vitamin C, and the concentration is 10 μg/mL (scale bar is 100 μm). Significance: ### *p* < 0.001 compared with the control group. ** *p* < 0.01, *** *p* < 0.001 compared with the H_2_O_2_ stimulation group. The data are shown as the mean ± S.D. (*n* = 5).

**Figure 11 molecules-27-01914-f011:**
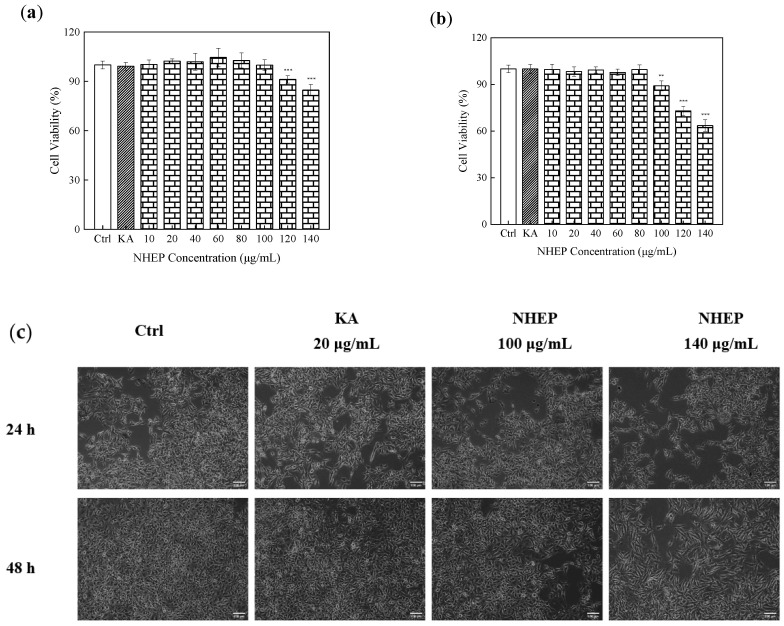
Cytotoxicity of NHEP on B16 cells: (**a**) viability of the B16 cells cultured with the NHEP for 24 h, (**b**) viability of the B16 cells cultured with the NHEP for 48 h, (**c**) cellular morphology of the B16 cells cultured with the NHEP for 24 h and 48 h (scale bar is 100 μm). Significance: compared with the control group, ** *p* < 0.01, *** *p* < 0.0001. The data shows as the mean ± S.D. (*n* = 5). KA stands for Kojic acid, and the concentration is 20 μg/mL.

**Figure 12 molecules-27-01914-f012:**
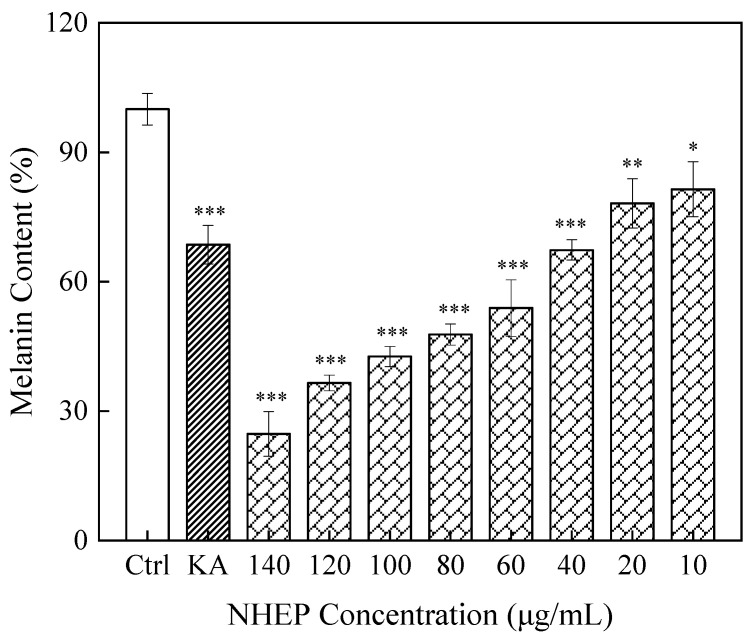
Effects of the NHEP on B16 cells melanin production. Significance: compared with the control group, * *p* < 0.05, ** *p* < 0.01, *** *p* < 0.001. The data shows as the mean ± S.D. (*n* = 5). KA stands for Kojic acid, and the concentration is 20 μg/mL.

**Table 1 molecules-27-01914-t001:** Scheme and experimental results of Box–Behnken.

NO.	A/(mL/g)	B/(%)	C/(%)	Flavonoids Content (mg/g)
1	40:1	4	50	26.62
2	40:1	5	60	25.66
3	30:1	3	50	21.30
4	40:1	5	40	23.34
5	40:1	3	40	22.83
6	50:1	4	60	22.73
7	40:1	4	50	26.40
8	40:1	4	50	27.06
9	30:1	4	60	22.30
10	50:1	4	40	22.65
11	30:1	4	40	20.67
12	40:1	4	50	26.92
13	30:1	5	50	23.02
14	50:1	3	50	22.85
15	40:1	4	50	26.95
16	50:1	5	50	22.72
17	40:1	3	60	23.52

**Table 2 molecules-27-01914-t002:** Variance analysis of Box–Behnken regression model results.

Source	Sum of Squares	df	Mean Squares	F-Value	*p*-Value
Model	71.7596	9	7.9733	67.6019	<0.0001 ***
A	1.6801	1	1.6801	14.2445	0.0069 **
B	2.2521	1	2.2521	19.0949	0.0033 **
C	2.7817	1	2.7817	23.5852	0.0018 **
AB	0.8540	1	0.8540	7.2411	0.0310 *
AC	0.5987	1	0.5987	5.0763	0.0589
BC	0.6680	1	0.6680	5.6636	0.0489 *
A^2^	38.6989	1	38.6989	328.1105	<0.0001 ***
B^2^	6.9468	1	6.9468	58.8986	0.0001 ***
C^2^	11.7205	1	11.7205	99.3726	<0.0001 ***
Residual	0.8256	7	0.1179		
Lack of Fit	0.5251	3	0.1750	2.3293	0.2159
Pure Error	0.3006	4	0.0751		
Cor Total	72.5852	16			
R^2^	0.9886				

R^2^ = Coefficients of determination. Significance: * *p* < 0.05, ** *p* < 0.01 and *** *p* < 0.001.

**Table 3 molecules-27-01914-t003:** The content of flavonoids and polyphenols in the NHE and NHEP with the same dry weight.

Sample	Compound	
	Flavonoids Content (mg/g)	Polyphenols Content (mg/g)
NHE	55.60 ± 0.38	414.97 ± 2.95
NHEP	157.39 ± 1.18 *	888.63 ± 4.41 *

The data shows as the mean ± S.D. (*n* = 3). *: *p* < 0.05 compared to NHE.

**Table 4 molecules-27-01914-t004:** Polyphenols and flavonoids compositions of NHEP by the characterization of HPLC.

Standard	Retention Time (min)	Regression Equation	R^2^	NHEP Compound (mg/g)
Gallic acid	5.33	y = 28.696x – 6.5794	0.9997	1.60 ± 0.36
Corilagin	16.21	y = 14.902x – 40.435	0.9992	65.60 ± 3.36
Ellagic acid	30.58	y = 10.3x – 11.978	0.9999	97.72 ± 0.92
Rutin	31.32	y = 10.71x – 9.2865	0.9999	34.41 ± 4.24
Myricetin	40.86	y = 12.76x + 4.3092	0.9996	3.13 ± 0.87
Quercetin	47.15	y = 11.001x + 2.6758	0.9999	4.64 ± 0.15
Naringin	50.69	y = 34.715x + 24.424	0.9999	4.03 ± 0.24

Y = The peak area (mAU*s); X = The concentration of standards (μg/mL); R^2^ = Coefficients of determination. The data shows as the mean ± S.D. (*n* = 3).

## Data Availability

Data are contained within the article.

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
