# Peer review of "Optimization of Ultrasound-Assisted Cellulase Extraction from Nymphaea hybrid Flower and Biological Activities: Antioxidant Activity, Protective Effect against ROS Oxidative Damage in HaCaT Cells and Inhibition of Melanin Production in B16 Cells"

_molecules, 2022, doi:10.3390/molecules27061914_

Round 1

Reviewer 1 Report

The study is interesting and provides new data foundations for the application of phenols from the Nymphaea hybrid Flower.

Line 65. microporous to macroporous.

Line 96. It looks like something is wrong with the liquid-to-solid ratio, 1:10 would be 10:1, same changes could be applied to all the others around this line.

Line 115. Significant differences should be marked on data in Figure 1.

Line 200. Significant differences should be marked on data in Figure 3.

Line 256 and 465, the volume and dimension (diameter* height) of the resin bed should be mentioned around both lines. The commonly used term was bed volume (BV), the flow speed can be converted to BV/ hour or min.

Line 234. Significant differences should be marked on data in Table 3.

Line 285. More discussion compared with the published phenolic compounds in Nymphaea hybrid Flower should be conducted. And there are many major peaks that are still unknown.  And what could they be?

Line 409. may due to may be due.

Line 465. Basic physical and chemical parameters could be provided here because the AB-8 is a local instead of an international brand. And what was the speed for crude extract loading absorption, washing, and ethanol desorption during dynamic resin operation?

Author Response

Response to Reviewer 1 Comments

Point 1: microporous to macroporous.

Response1: Thank you for your correction.

[In line 65 of the revised manuscript, change “microporous” to “macroporous”.]

Point 2: It looks like something is wrong with the liquid-to-solid ratio, 1:10 would be 10:1, same changes could be applied to all the others around this line.

Response 2: Thank you for pointing out the mistakes in writing. The reviewer is correct, and we have modified it in the revised manuscript.

[In Figure 1a and 3, Table 1, line 96-98, 147, 164, 441 and 563 of the revised manuscript, change “1:20” to “20:1”, “1:40” to “40:1” and “1:60” to “60:1”.]

Point 3: Significant differences should be marked on data in Figure 1.

Response 3: Thanks for your advice. We seriously modified it.

[In Figure 1 and line 117 of the revised manuscript, we analyzed the significance between groups, *: p < 0.05 compared between groups.]

Point 4: Significant differences should be marked on data in Figure 4.

Response 4: Thanks for your advice. We have made the marks.

[In Figure 3 and line 203 of the revised manuscript, we analyzed the significance between groups, *: p < 0.05 compared between groups.]

Point 5: The volume and dimension (diameter* height) of the resin bed should be mentioned around both lines. The commonly used term was bed volume (BV), the flow speed can be converted to BV/ hour or min.

Response 5: Thanks for your comments, we seriously corrected the mistakes.

[In lines 213-221, 480-482 and Figure 4b of the revised manuscript, we measured the volume diameter length of the resin bed is 22 and 300 mm and calculated the volume of the resin bed (BV) is 25mL. We changed “0.5 mL/min” to “1.2 BV/h”, “1 mL/min” to “2.4 BV/h”, “2 mL/min” to “4.8 BV/h”, “3 mL/min” to “7.2 BV/h”.]

Point 6: Significant differences should be marked on data in Table 3.

Response 6: Thanks for your advice. We seriously modified it.

[In Table 3 and line 237 of the revised manuscript, we analyzed the significance between groups, *: p < 0.05 compared to NHE.]

Point 7: More discussion compared with the published phenolic compounds in Nymphaea hybrid Flower should be conducted. And there are many major peaks that are still unknown. And what could they be?

Response 7: Thanks to the reviewers for their professional and serious suggestions. It’s meaningful. We added some references to compare with some standards in the NHEP. In the HPLC experiment, we tried to use different standard compounds to identify every peak according to the research reports of other Nymphaea plants, but failed, only seven compounds in the NHEP were identified. On the other hand, the purpose of the paper is mainly focused on the extraction from Nymphaea hybrid, purification and evaluation of antioxidant and whitening capacity in vitro. In the future, we may take a more in-depth analysis of the structure of NHEP by the HPLC-MS method.

[In lines 286-293 of the revised manuscript, Uddin et al. identified 9 bioactive phytochemicals from methanolic extract of Nymphaea nouchali using an HPLC system [54]. Among these compounds, rutin (39.44 mg/g), myricetin (30.77 mg/g), ellagic acid (11.05 mg/g), gallic acid (5.33 mg/g) and quercetin (0.90 mg/g) were polyphenols or flavonoids, which were also identified as bioactive ingredients of NHEP in this work. We found the ellagic acid content (97.72 ± 0.92 mg/g) of NHEP was much greater than that (11.05 mg/g), of Nymphaea nouchali and that (6.16 ± 0.019 mg/g), of ethanolic extract of Water Lily (Nymphaea Tetragona Georgi) [55].]

Point 8: may due to may be due.

Response 8: Thanks for your correction.

[In line 413 of the revised manuscript, change “may due” to “may be due”]

Point 9: Basic physical and chemical parameters could be provided here because the AB-8 is a local instead of an international brand. And what was the speed for crude extract loading absorption, washing, and ethanol desorption during dynamic resin operation?

Response 9: Thanks for your comments. We supplemented the basic physical and chemical parameters of AB-8; and supplemented the sample concentration, pH ethanol concentration on static adsorption; and supplemented the quality of resin, sample volume, and sample flow rate on dynamic desorption.

[In lines 473-475 of the revised manuscript, AB-8 macroporous resins are weak polar resins, milk-white spherical particles, and the specific surface area is 480 ~ 520 m2/g.]

[In lines 481-485 of the revised manuscript, adsorption of concentration of NHE (0.33 ~1.15 mg/mL), pH (2 ~ 6) and desorption of ethanol concentration (50 ~ 90%) on the static adsorption and desorption experiments, and sample volume (120 mL) and sample flow rate (1.2 ~ 7.2 BV/h) on the dynamic experiments. Dynamic experiments: Chromatographic columns: diameter × length (22 × 300 mm); the quality of the resin was 5 g; the volume of the resin bed (BV) was 25 mL.]

Reviewer 2 Report

Dear Authors,

Your study involving:

Optimization of Ultrasound-assisted Cellulase Extraction from 2
Nymphaea hybrid Flower and Biological Activities: Antioxidant 3
Activity, Protective Effect Against ROS Oxidative Damage in 4
HaCaT Cells and Inhibition of Melanin Production in B16 Cell

is a very well executed project with application of ultrasound-assisted cellulase extraction. As this plant has medicinal value, cosmetics application and ideal method of extraction which does not degrade, oxidize the key components of the biomedical value is important. I do have few concerns.

for example,

All the figure legends were very small font, can you increase the font size to 10. It was hard to read.

  • The figure 6 has alcohol, aromatics, and carbonyl funtional groups. We also see some C-H stretching and bending as per the IR report. Can these be discussed in the discussion?
  • Were any NMR studies performed on the crude mixture. It might help to the reader to what type of compounds are present in this extract and compare with the literature reported NMR studies.

The antioxidant potential, biological cell viability assays support the medicinal value but with out the structures it is hard to make analogs and perform SAR studies.

Good work.

Author Response

Response to Reviewer 2 Comments

Point 1: All the figure legends were very small font, can you increase the font size to 10. It was hard to read.

Response 1: Thanks for your advice. We seriously modified it.

[In Figure 1-12 of the revised manuscript, we modified the font size of these Figures.]

Point 2: The figure 6 has alcohol, aromatics, and carbonyl funtional groups. We also see some C-H stretching and bending as per the IR report. Can these be discussed in the discussion?

Response 2: Thanks for your advice. we have a simple analysis for C-H stretching and bending.  

[In lines 272-274 of the revised manuscript: The adsorption peaks at 1449 and 1349 cm-1 are assigned to the bending vibrations of −CH3 and −CH2. The peaks at 1210 and 1038 cm-1 wavelength belong to the stretching vibration of C−H and C−O bonds]

Point 3: Were any NMR studies performed on the crude mixture. It might help to the reader to what type of compounds are present in this extract and compare with the literature reported NMR studies.The antioxidant potential, biological cell viability assays support the medicinal value but with out the structures it is hard to make analogs and perform SAR studies

Response 3: Thanks for your good advice. We have characterized the extract NHEP using IR and HPLC method, no NMR study was carried out in this paper. IR can identify some functional groups of the mixture. In the HPLC experiments, we tried to use different standard compounds with determined structures to identify every peak according to the research reports of other Nymphaea plants, and finally seven compounds in the NHEP were identified. NMR is a good method to identify the structure of ingredients, but there were few reports about NMR studies on the crude extract of other Nymphaea plants. In the future, we hope more in-depth analysis of the structure of NHEP by different methods and provide support to make analogs and perform SAR studies to study the medicinal value.